# An Updated Overview of MRI Injuries in Neonatal Encephalopathy: LyTONEPAL Cohort

**DOI:** 10.3390/children9040561

**Published:** 2022-04-14

**Authors:** Jonathan Beck, Gauthier Loron, Pierre-Yves Ancel, Marianne Alison, Lucie Hertz Pannier, Philippe Vo Van, Thierry Debillon, Nathalie Bednarek

**Affiliations:** 1Department of Neonatology, Reims University Hospital Alix de Champagne, 51100 Reims, France; gloron@chu-reims.fr (G.L.); nbednarek@chu-reims.fr (N.B.); 2EPOPé (Obstetrical Perinatal and Pediatric Epidemiology Research Team), CRESS (Centre of Research in Epidemiology and StatisticS), INSERM (Institut National de la Santé et de la Recherche Médicale), INRAE (Institut National de la Recherche Agronomique), Université de Paris, 75004 Paris, France; pierre-yves.ancel@inserm.fr; 3CReSTIC EA (Centre de Recherche en Traitement du Signal Informatique) 3804, Université de Reims Champagne Ardenne, 51097 Reims, France; 4Assistance Publique—Hôpitaux de Paris, Clinical Investigation Center P1419, 75004 Paris, France; 5Service d’Imagerie Pédiatrique, Hôpital Robert Debré, APHP (Assistance Publique—Hôpitaux de Paris), 75019 Paris, France; marianne.alison@aphp.fr; 6Unit 1141 NeuroDiderot, Inserm, CEA (Commissariat à l’Énergie Atomique et aux Énergies Alternatives), Université Paris Cité, 75019 Paris, France; lucie.hertz-pannier@cea.fr; 7UNIACT (Unité de Recherche en NeuroImagerie Applicative Clinique et Translationnelle), Neurospin, CEA (Commissariat à l’Énergie Atomique et aux Énergies Alternatives)-Saclay, 91191 Gif sur Yvette, France; 8Department of Neonatology, Hospices Civils de Lyon, Femme Mère Enfant Hospital, Pinel, 69500 Bron, France; philippe.vo-van@chu-lyon.fr; 9Neonatal Intensive Care Unit CHU (Centre Hospital-Universitaire) Grenoble Alpes, 38000 Grenoble, France; tdebillon@chu-grenoble.fr; 10Grenoble INP (Institut d’Ingénierie et de Management), TIMC (Techniques de l’Ingénierie Médicale et de la Complexité)-IMAG (Informatique, Mathématiques et Applications, Grenoble), CNRS (Centre National de la Recherche Scientifique), University Grenoble Alpes, 38000 Grenoble, France

**Keywords:** hypoxic−ischemic encephalopathy, magnetic resonance imaging, brain injury, newborn

## Abstract

Background: Brain magnetic resonance imaging (MRI) is a key tool for the prognostication of encephalic newborns in the context of hypoxic−ischemic events. The purpose of this study was to finely characterize brain injuries in this context. Methods: We provided a complete, descriptive analysis of the brain MRIs of infants included in the French national, multicentric cohort LyTONEPAL. Results: Among 794 eligible infants, 520 (65.5%) with MRI before 12 days of life, grade II or III encephalopathy and gestational age ≥36 weeks were included. Half of the population had a brain injury (52.4%); MRIs were acquired before 6 days of life among 247 (47.5%) newborns. The basal ganglia (BGT), white matter (WM) and cortex were the three predominant sites of injuries, affecting 33.8% (*n* = 171), 33.5% (*n* = 166) and 25.6% (*n* = 128) of participants, respectively. The thalamus and the periventricular WM were the predominant sublocations. The BGT, posterior limb internal capsule, brainstem and cortical injuries appeared more frequently in the early MRI group than in the late MRI group. Conclusion: This study described an overview of brain injuries in hypoxic−ischemic neonatal encephalopathy. The basal ganglia with the thalamus and the WM with periventricular sublocation injuries were predominant. Comprehensive identification of brain injuries in the context of HIE may provide insight into the mechanism and time of occurrence.

## 1. Introduction

Neonatal encephalopathy, related to a mechanism of peripartum asphyxia, usually named hypoxic−ischemic encephalopathy (HIE), may lead to severe neurological impairment in children and remains a public health issue [1].

Brain magnetic resonance imaging (MRI) is a key tool for prognostication in this vulnerable population.

A full-term newborn’s brain exhibits areas that are highly vulnerable to hypoxic−ischemic events, such as the basal ganglia thalami (BGT), cortex and white matter (WM) [2,3]. These structures are broadly involved in motor processing, behavior and cognition development [2,3].

Few studies have reported an exhaustive characterization of anatomical injuries in the context of HIE and their incidence.

MRI classifications, which are widely used, allow a quantitative analysis, but give little information about the incidence of brain injuries and the individual outcome (i.e., the same score may reflect different patterns of injury) [2,4,5,6,7]. A few studies reported more anecdotical brain injuries involving the corpus callosum, brainstem, cerebellum and their evolution [8,9,10].

Establishing a prognostic correlation between anatomical injuries and future functioning remains relevant to target neuroprotection, remediation and rehabilitation.

At the present time, therapeutic hypothermia (TH), applied within 6 h after birth for 72 h, reduces death and disability [11].

Classically, brain injuries appear progressively during the first hours to days following the anoxic–ischemic attack on diffusion-weighted imaging (DWI); the classical sequences are relevant from about the tenth day. Even if hypothermia seems not to alter the prognostic accuracy of brain imaging, MRI characterization was mostly established before the era of TH [11,12].

Brain cooling influences the time course of the MRI signal on both conventional and diffusion sequences [11,12]. Indeed, therapeutic hypothermia introduces a delay of 2–4 days (D) in the pseudonormalization of DWI [12]. There are little data on whether this effect applies differently to different brain areas.

This study aims to characterize and report the incidence of brain injuries in the context of neonatal encephalopathy in full-term newborns in the large, national cohort LyTONEPAL (Long-Term Outcome of Neonatal hypoxic EncePhALopathy in the era of neuroprotective treatment with hypothermia) [13].

## 2. Materials and Methods

### 2.1. Population

The LyTONEPAL observational, prospective cohort included newborns with moderate and severe neonatal encephalopathy born at ≥36 wg between September 2015 and March 2017. Sixty-eight (out of a total of sixty-nine) French centers participated in this study.

The other inclusion criteria were defined according to clinical and biological data, detailed in the study protocol of LyTONEPAL cohort [13].

The criteria for non-inclusion were: infants with congenital malformations, chromosomal disorders, congenital neuromuscular disorders and traumatic birth not adhering to HIE criteria [13].

### 2.2. Data Collection

Clinical data included medical history of delivery, maternity level, demographical and biometrical data of the newborn, resuscitation at birth, Apgar score, neurological evaluation and grade of encephalopathy [14], time to reach 34 °C and clinical seizures. Sentinel event was defined by the occurrence of a cord prolapse, a head retention, a retroplacental hematoma, a uterine rupture, a cord rupture, an amniotic embolism, a fetal−maternal hemorrhage or a maternal shock. Grade of encephalopathy was evaluated by Sarnat & Sarnat score, which assesses consciousness, tone, reflexes, pupillary reactivity, oculomotor functionality, sucking and the presence/absence of seizures.

Biological variables included first-hour acid–base balance (pH), lactate values and blood glucose values.

A standardized reading form was provided to the senior radiologist of each cooling center to help with the interpretation of brain MRIs (Appendix A). It was created by an expert panel of French radiologists.

Seven major brain areas were characterized on DWI and conventional (T1/T2) sequences on MRI performed during the first twelve days of life: basal ganglia thalami, white matter, cortex, posterior internal limb capsule (PLIC), corpus callosum (CC), cerebral stem (brainstem) and cerebellum.

Six of these seven brain areas were further detailed by their sublocations.

Basal ganglia thalami (BGT) were distinguished as thalami, globus pallidus, putamen and caudate nucleus. White matter (WM) was distinguished as periventricular, parietal, frontal, centrum semiovale, temporal and occipital. Cortical areas were distinguished as rolandic, posterior junctional, anterior junctional, occipital, mesiotemporal and insula. The splenium and knee were distinguished for the corpus callosum and the midbrain; pons and medulla were distinguished for the brainstem.

On the other hand, radiologists were encouraged to describe the other types of injuries they observed other than those described in the manuscript. Patients with isolated vascular lesions, such as sinus thrombosis or stroke, were excluded. Traumatic injuries associated with hypoxic−ischemic criteria were considered if described by radiologist.

For infants who underwent more than one MRI, the scan with the worst pattern was considered. When results were concordant, the late MRI was selected. The population was divided into 2 groups—those with MRI before D6 (early MRI) and those with MRI between D6-D12 (late MRI)—and MRI findings, and patient characteristics were compared. The radiological examinations realized after 2 weeks of life were not taken into account because of the differences in brain maturation and expected information.

### 2.3. Data Management and Statistics

Frequency of injury was reported for each considered sublocation for each group according to their hypothermic or normothermic status.

Descriptive analyses of qualitative data were expressed as *n* (%), and quantitative data were expressed as mean ± standard deviation.

Statistical analyses involved using Intercooled STATA v16 (Stata Corp., College Station, TX, USA).

## 3. Results

### 3.1. Population

Among 794 newborns enrolled in LyTONEPAL, 520 patients were born ≥36 wg (479 patients treated by TH) and had at least one available MRI between D0 and D12 (Figure 1).

Characteristics of the population are presented in Table 1.

Newborns under hypothermic conditions were mainly born outside a TH center (72.5%). Of note, a sentinel event was observed for more than half of the cohort (50.8%), and a fetal heart rate abnormality was seen in a large part of the cohort (85%). A large part of the cohort required intubation in the delivery room (75.1%). Neonatal encephalopathy was moderate for two-thirds of the cohort (63.5% grade II). Newborns had a good short-term outcome with a large rate of normal neurological exams at discharge (72.4%). The death rate during hospitalization was low (15.6%).

For the 41 newborns in normothermic condition (7.9%), we observed that they seemed to demonstrate better neonatal adaptation with a lower Apgar score > 5 at 5 min (*n* = 15, 36.6%) and less intubation in the delivery room (*n* = 20, 48.8%). However, for these newborns, we observed more seizures during the first week of life (*n* = 23, 56.1%), notably before 24 h of life (n = 18, 43.9%). Regarding the other characteristics, the newborns without TH appeared to be similar to those with TH.

### 3.2. Brain Injuries in Neonatal Encephalopathy

Overview of brain injuries in neonatal encephalopathy is describe in Table 2.

Two hundred seventy-three newborns (52.5%) had brain MRI injuries. The cortex, WM and BGT were the three predominant sites of injuries, affecting 25.6% (*n* = 128), 33.5% (*n* = 166) and 33.8% (*n* = 171) of the whole population, respectively.

The thalamus was the most involved sublocation for the BGT injuries group (70.7%, *n* = 121). PLIC injuries were observed for 82 newborns (16.5% of the whole population, 48% of the BGT injuries group). Periventricular WM injuries were the most observed sublocation in the whole population. CC, brainstem and cerebellum injuries were observed in 63 (13.7%), 44 (8.8%) and 24 (4.9%) newborns, respectively. Some pictures of cerebral injuries on MRI are presented in Figure 2.

### 3.3. Hypothermia and Brain Injuries in Neonatal Encephalopathy

Four hundred seventy-nine newborns received TH (92.1%); the rate and repartition of brain injuries were similar to those of the whole population (Table 2).

Of the 41 newborns who did not receive TH, 25 had brain injuries (61.0%). BGT, WM and cortical injuries were observed in 14 (35.0%), 18 (45.0%) and 14 (34.2%) newborns, respectively. The thalamus was also the most involved sublocation for the BGT injuries group (85.7%, *n* = 12). PLIC injuries were observed for 10 newborns (24.4% of the 41 non-TH newborns, 71.4% of the BGT injuries group). There was no predominant sublocation for the WM or the cortex. Corpus callosum, brainstem and cerebellum injuries were observed in 7 (18.4%), 5 (12.5%) and 1 (2.4%) newborns, respectively.

### 3.4. The Influence of MRI Timing in Identifying Brain Injury

Early MRI (before D6) was performed for 247 newborns (47.5%). Late MRI (D6 to D12) was performed for 273 newborns (52.5%). Delivery and newborn characteristics were similar between the two groups (early or late MRI) (Table 3).

Acute perinatal events, abnormalities of cardiac fetal rate, severe NE and death were more frequent in the early MRI group than in the late MRI group.

BGT, PLIC, brainstem and cortical injuries appeared more frequently in the early MRI group than in the late MRI group (Table 4).

Of 520 patients with available MRIs, 484 patients had a single MRI, and 36 patients had 2 MRIs. Thirty-two (88.9%) patients had similar results: thirteen normal and nineteen pathological MRIs. For four patients, of which three were on TH, the analyses were not concordant. For patient 1 undergoing TH, the initial MRI (D4) showed diffuse WM injury (T1 and T2 sequences), whereas the second MRI’s (D9) interpretation was normal (T1 and T2 sequences). For patient 2 undergoing TH, the first MRI (D5) interpretation was normal (DWI, T1 and T2 sequences), whereas the second MRI (D7) showed moderate BGT and WM injuries (DWI, T1 and T2 sequences). For patient 3 undergoing TH, the first MRI interpretation (D3) was normal (DWI and T2 sequences), whereas the second MRI (D8) showed moderate WM and cortical injuries (T1 and T2 sequences). For patient 4 (not undergoing TH), the first MRI (Day 1) showed moderate WM injury (in DWI), whereas the second MRI (D8) was normal (T1 and T2 sequences).

## 4. Discussion

The LyTONEPAL project updated brain injury locations and incidence in the context of neonatal encephalopathy in a large, prospective cohort. Brain injuries due to hypoxic−ischemic events are diffuse and do not spare any region. Basal ganglia, especially the thalami, white matter and cortical injuries, were predominant. Early MRI seemed to identify more brain injuries within the first week of life. The characteristics of this present cohort are consistent with other cohorts dealing with neonatal encephalopathy due to hypoxic−ischemic events: unexpected pathology and the observation of perinatal events (sentinel events, fetal heart rhythm abnormalities, poor adaptation to the extrauterine life and a large proportion of mild encephalopathy) [5,15,16].

### 4.1. Brain Injuries’ Location and Literature

Previous work on LyTONEPAL MRIs reported HIE as a diffuse brain pathology, in agreement with the literature [16,17,18,19,20]. It was also observed that PLIC and brainstem injuries were mainly associated with BGT [17]. Our study supported this finding and characterized the sublocations of these major brain areas and their incidence.

BGT involvement was not limited to thalamic lesions. Although these were predominant, the other parts of the BGT (putamen, globus pallidus, caudate nucleus) were also damaged in lesser proportions. A few studies with a smaller number of patients described BGT and WM brain injuries in the same way [21,22,23,24,25].

The three major locations, namely the BGT, the cortex and the white matter, as well as the PLIC, have been integrated into several classifications for the assessment of prognosis [2,4,5,6,7]. In these classifications, the BGT were either globally integrated, detailed with the thalamus and lentiform nucleus [2,3,4,7] or detailed with the putamen, caudate nucleus, globus pallidus and thalamus [12,26].

The purpose of these classifications is to assess the severity of the brain injuries’ locations [3,4,12].

However, characterization of the areas involved gives valuable indications to understand the mechanism and timing of the injury. In the near future, this might be crucial for testing adjunctive therapies for hypothermia. Finally, it is essential for individual prediction of prognosis.

### 4.2. Brain Injury and TH

Since TH initiation began as a neuroprotective treatment for perinatal HIE, a significant decrease in brain injuries has been described [3,6,18]. Shankaran et al. and Rutherford et al. observed more normal MRIs in patients who underwent TH, and this was associated with a decrease in BGT, PLIC and WM injuries [3,6]. Inder et al. observed a decrease in cortical and WM injuries and isolated BGT injuries [18]. Brain volumes were also impacted, with better cortical volumes demonstrated for neonates treated by TH [27].

In comparison with previous studies, this present study highlights the positive impact of therapeutic hypothermia through the observation of a higher rate of normal MRIs [3,4,19,28,29,30,31], a lower or stable rate of severe lesions (e.g., in the BGT) [3,16,19,28,29,30,31] and a lower rate of cortical injuries [3,16,19,29,30,31].

The small number of normothermic patients, as well as possible interfering factors (infection, intrauterine growth retardation, etc.), lead us to remain cautious in every interpretation of our observations.

### 4.3. Early or Late MRI?

The optimal MRI timing for NE patients is still discussed in the literature [2,3,32,33]. The gold standard for brain injury evaluation was examining T1/T2 sequences during the second week of life [27,33]. With the implementation of DWI sequences in the era of TH, MRI is performed during the first week after birth. Severe outcomes (Sarnat grade III NE and death) and severe brain injury (BGT/PLIC injuries) were more frequently observed in the early MRI group. We hypothesized that newborns with severe injuries would benefit from an earlier brain assessment for ethical reasons. Diffusion sequences obtained during the first six days after birth reflect conventional sequences T1/T2 obtained during the second week after birth [33,34]. For half of the population, brain injury was assessed once by early MRI with DWI sequences. An early MRI between 24 and 96 h of life to delineate the timing of perinatal cerebral injury and a late MRI between 7 and 21 days of life to delineate brain injuries were previously recommended [34]. Early MRI with DWI sequences may help manage patients in the subacute phase [33,34,35]. Of note, in this large cohort, more brain injuries were identified within the first six days of life, and few differences were observed between early and late MRI brain injury identification for patients who were scanned twice.

### 4.4. Perspectives

One of the objectives of the LyTONEPAL study is to evaluate the relationship between anatomical structures and their function: in other words, to evaluate the motor and cognitive outcomes of the children in the cohort in relation to their brain injuries.

The final goal is to evaluate the neurological prognosis very early and accurately and to be able to propose individualized motor rehabilitation and cognitive mediation in a period of propitious plasticity.

### 4.5. Strengths and Limitations

One of the strengths of this study is its representativeness of the recent French experience in the evaluation of brain injuries in neonatal encephalopathy. The quality of the MRI data, although without centralized reading, was guaranteed by the standardized MRI protocol and reporting and the experience of the radiologists in French TH reference centers. However, despite a standardized MRI protocol, in the absence of a centralized review, each item has not been filled (explaining the variation of the denominators in Table 2); the impact is related to the quality of the MRI and the filling of the data by another person other than the radiologist himself.

We did not consider the type of MRI sequence where the injury was visible. Traumatic brain injuries have certainly been underestimated because of the design of this study. However, these types of lesions are often associated with instrumental maneuvers made necessary by the fetal condition (fetal heart rhythm abnormalities). It remains particularly difficult in this context to distinguish the part due to traumatic birth injuries from the hypoxic−ischemic phenomenon, often associated with cases of neonatal encephalopathy [36]. This study provided extended data on injured newborns according to their therapeutic conditions and an updated and detailed overview of brain injuries in the context of HIE.

## 5. Conclusions

This work detailed brain injuries in the context of neonatal encephalopathy due to hypoxic−ischemic events. Basal ganglia injuries, especially those involving the thalami, and white matter injuries within the periventricular location were predominant. Comprehensive identification of brain injuries in the context of HIE may provide insight into the mechanisms involved and their time of occurrence. Confrontation of anatomical injuries observed in the neonatal period remains necessary to improve prognostication, which aims to deliver precise information for parents and personalize the future care of the infant.

## Figures and Tables

**Figure 1 children-09-00561-f001:**
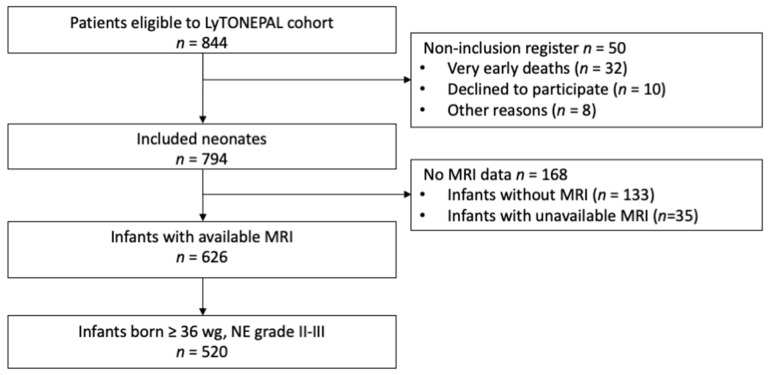
Flowchart. Abbreviations: WG = weeks’ gestational, NE = neonatal encephalopathy, grade I–II–III = stage of encephalopathy according to Sarnat classification [14], D = day.

**Figure 2 children-09-00561-f002:**
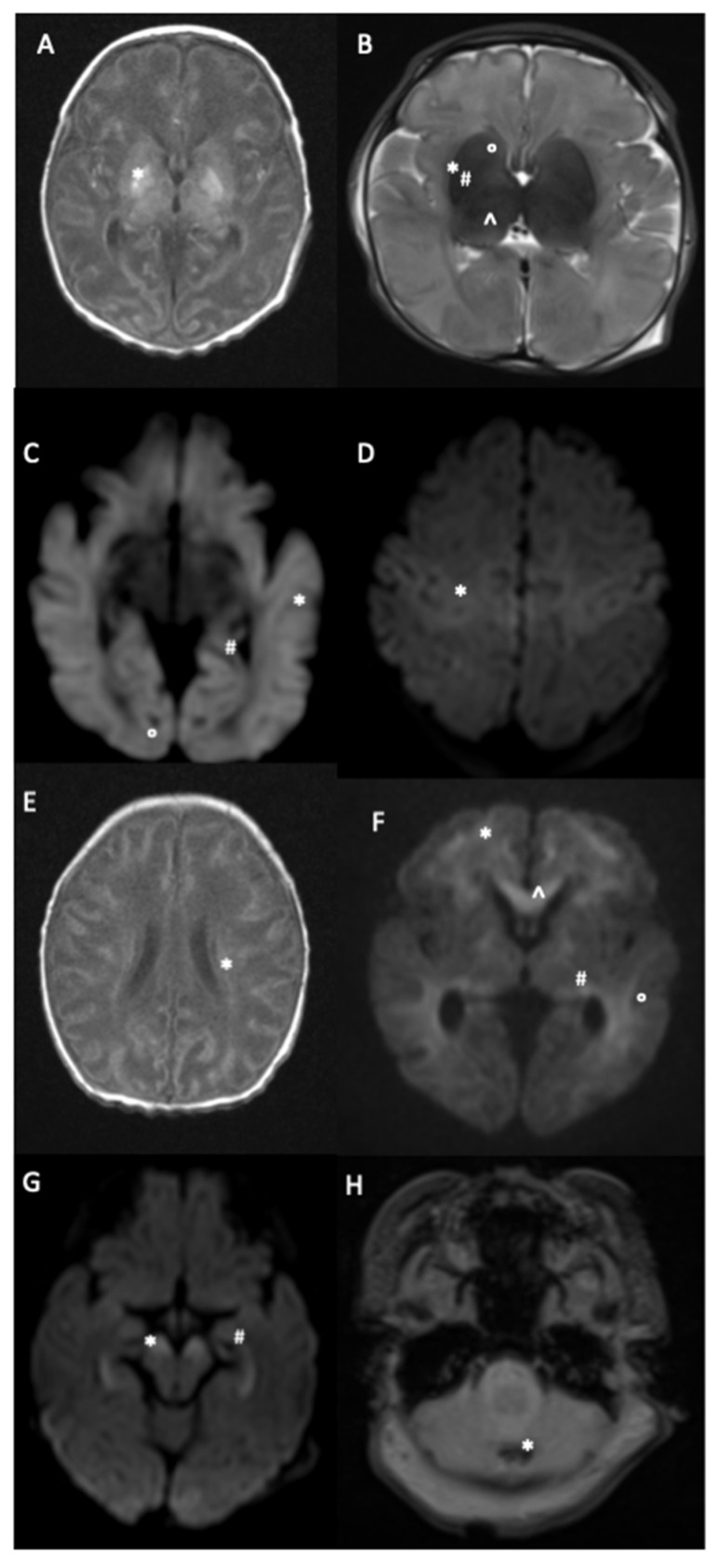
MRI brain injuries in context of neonatal encephalopathy. (**A**) = Axial T1 weighted image (T1W): high signal intensity (SI) of the diffuse cortex and the bilateral basal ganglia involving the globus pallidus (*). (**B**) = Axial 2 weighted image (TW2). Extensive high SI of the white matter and heterogeneous SI of the basal ganglia involving the caudate nucleus (°), putamen (*), globus pallidus (#), and thalamus (^). (**C**) = Diffusion weighted image (DWI): extensive restriction diffusion of the white matter and cortex including the external temporal (*), internal temporal (#) and occipital (°) lobes. (**D**) = DWI: bilateral restriction diffusion of the Rolandic sulci (*). (**E**) = Axial T1W: diffuse high SI of the cortex and centrum semi ovale. (*) (**F**) = DWI: diffuse SI of the anterior watershed region (*) the parietal white matter (°), the PLIC (#) and corpus callosum (^). (**G**) = DWI: restriction signal of the peduncular brain stem (*) and the hippocampus (#). (**H**) = T2W gradient echo: punctiform cerebellum injury in low SI (*).

**Table 1 children-09-00561-t001:** Characteristics of the population.

Newborn ≥ 36 wg, NE Grades II–III		
Patient Characteristics	All *n* = 520	With TH *n* = 479
	*n* (%) Mean ± SD	*n* Group (%) Mean ± SD
Birth outside a TH center	377/520 (72.5)	349/479 (72.9)
Sentinel event	264/520 (50.8)	248/479 (51.8)
Abnormalities of fetal heart rate	436/513 (85.0)	400/472 (84.7)
Delivery mode	511/520	470/479
Vaginal, no instrumental extraction	89 (17.4)	78 (16.6)
Vaginal, instrumental extraction	122 (21.9)	103 (21.9)
Cesarean	310 (60.7)	289 (61.5)
Term, WG	39.4 ± 1.5	39.4 ± 1.6
Birth weight, g	3172 ± 537	3180 ± 523
Sex, male	278/520 (53.5)	255/479 (53.2)
Apgar score at 5 min < 5	283/520 (54.4)	268/479 (55.9)
Apgar score at 10 min < 5	219/439 (49.9)	207/403 (51.4)
Intubation in delivery room	388/517 (75.1)	368/476 (77.3)
Encephalopathy grade *(Sarnat)* ^a^	520/520	479/479
II	330 (63.5)	304 (63.6)
III	190 (36.5)	175 (36.5)
First-hour pH	6.97 ± 0.18	6.96 ± 0.18
First-hour lactate (mmol/L)	12.43 ± 4.96	12.50 ± 4.97
First-hour base excess (mmol/L)	11.81 ± 6.64	11.93 ± 6.64
Glycemia at admission (mmol/L)	6.87 ± 4.59	6.95 ± 4.67
Hypoglycemia ≤ 24 h of life ^b^	36/435 (8.3)	35/400 (8.8)
Seizures ≤ 24 h of life	145/514 (28.2)	127/473 (26.9)
Seizures > 24 h of life	99/514 (19.3)	91/473 (19.2)
Seizures during the first 8 days of life	190/520 (36.5)	167/479 (34.9)
Normal clinical exam at discharge	276/381 (72.4)	257/351 (73.2)
Death during hospitalization	81/520 (15.6)	74/479 (15.4)

Abbreviations: WG = weeks’ gestation, TH = therapeutic hypothermia. ^a^ Sarnat & Sarnat grade (1976) [14]. ^b^ Hypoglycemia defined by glucose level < 2.2 mmol/L.

**Table 2 children-09-00561-t002:** Brain injuries in neonatal encephalopathy, with or without therapeutic hypothermia.

Newborn ≥ 36 wg, NE Grades II–III			
Location	All (*n* = 520) *n* (%)	with TH (*n* = 479) *n* (%)	without TH (*n* = 41) *n* (%)
Any location	273/520 (52.5)	248/479 (51.8)	25/41 (60)
BGT	171/506 (33.8)	157/466 (33.7)	14/40 (35.0)
- Thalamus	121	109	12
- Globus pallidus	93	84	9
- Putamen	85	78	7
- Caudate nucleus	59	52	7
PLIC	82/497 (16.5)	72/456 (15.8)	10/41 (24.4)
WM	166/496 (33.5)	148/456 (32.5)	18/40 (45.0)
- Periventricular	68	62	6
- Junctional			
◦ anterior	49	43	6
◦ posterior	48	42	6
- Frontal	64	57	7
- Parietal	62	57	5
- Centrum semiovale	57	52	5
- Temporal	43	39	4
- Occipital	43	36	7
Cortex	128/500 (25.6)	114/459 (24.8)	14/41 (34.2)
- Rolandic	62	55	7
- Junctional			
◦ posterior	45	41	4
◦ anterior	43	40	3
- Occipital	58	53	5
- Mesiotemporal	48	45	3
- Insula	37	34	3
CC	63/460 (13.7)	56/422 (13.3)	7/38 (18.4)
- Splenium	57	52	5
- Knee	40	39	1
Brainstem	44/501 (8.8)	39/461 (8.5)	5/40 (12.5)
- Midbrain	28	25	3
- Pons	26	24	2
- Medulla	17	17	0
Cerebellum	24/491 (4.9)	23/450 (5.1)	1/41 (2.4)
- Hemisphere	15	14	1
- Vermis	5	5	0

Abbreviations: NE = neonatal encephalopathy, grade I–II–III = stage of encephalopathy according to Sarnat classification [14], TH = therapeutic hypothermia, BGT = basal ganglia thalami, CC = corpus callosum, PLIC = posterior-limb internal capsule, WM = white matter, D = day.

**Table 3 children-09-00561-t003:** Characteristics of the population according to the timing of MRI.

Patient Characteristics	All Newborns ≥36 wg *n* = 520	Early MRI (before D6) *n* = 247	Late MRI (between D6 and D12) *n* = 273
	*n* (%) Mean ± SD	*n* (%) Mean ± SD	*n* (%) Mean ± SD
Birth outside a TH center	377/520 (72.5)	181/247 (73.3)	196/273 (71.8)
Sentinel event	264/520 (50.8)	137/247 (55.5)	127/273 (46.5)
Abnormalities of fetal heart rate	436/513 (85.0)	210/242 (86.8)	226/271 (83.4)
Delivery mode	511/520	243/244	268/273
Vaginal, no instrumental extraction	89 (17.4)	38 (15.6)	51 (19.0)
Vaginal, instrumental extraction	112 (21.9)	54 (22.2)	58 (21.6)
Cesarean	310 (60.7)	151 (62.2)	159 (59.4)
Term, WG	39.4 ± 1.5	39.5 ± 1.6	39.4 ± 1.5
Birth weight, g	3172 ± 537	3167 ± 534	3177 ± 540
Sex, male	278/520 (53.5)	139/247 (56.3)	139/273 (50.9)
Apgar score at 5 min < 5	283/520 (54.4)	139/247 (56.3)	144/273 (52.8)
Apgar score at 10 min < 5	219/439 (49.9)	104/208 (50.0)	115/231 (49.8)
Intubation in delivery room	388/517 (75.1)	182/245 (74.3)	106/272 (75.7)
Encephalopathy grade (Sarnat) ^a^	520/520	244/244	273/273
II	330 (63.5)	147 (59.5)	183 (67.0)
III	190 (36.5)	100 (40.5)	90 (33.0)
First-hour pH	6.97 ± 0.18	6.95 ± 0.18	6.98 ± 0.19
First-hour lactate (mmol/L)	12.43 ± 4.97	12.48 ± 5.27	12.38 ± 4.68
First-hour base excess (mmol/L)	11.81 ± 6.64	12.12 ± 6.84	11.54 ± 6.46
Glycemia at admission (mmol/L)	6.87 ± 4.59	6.81 ± 4.76	6.93 ± 4.45
Hypoglycemia ≤ 24 h of life ^b^	36/435 (8.3)	18/209 (8.6)	18/226 (8.0)
Seizures ≤ 24 h of life	145/514 (28.2)	77/243 (31.7)	68/271 (25.1)
Seizures > 24 h of life	99/514 (19.3)	49/242 (20.3)	50/272 (18.4)
Seizures during the first 8 days of life	190/520 (36.5)	95/247 (38.5)	95/273 (34.8)
Normal clinical exam at discharge	276/381 (72.4)	123/166 (74.1)	153/215 (71.2)
Death during hospitalization	81/520 (15.6)	49/247 (19.8)	32/273 (11.7)

Abbreviations: WG = weeks’ gestation, TH = therapeutic hypothermia. ^a^ Sarnat & Sarnat grade (1976) [14]. ^b^ Hypoglycemia defined by glucose level < 2.2 mmol/L.

**Table 4 children-09-00561-t004:** Brain injuries in newborns with neonatal encephalopathy by date of MRI.

Brain Injuries	Total *n* = 520	Early MRI (before D6) *n* = 247	Late MRI (between D6 and D12) *n* = 273
	*n* (%)	*n* (%)	*n* (%)
No injury	247/520 (47.5)	106/247 (42.9)	141/273 (51.7)
BGT	171/506 (33.8)	94/236 (39.8)	77/270 (28.5)
WM	166/496 (33.5)	85/231 (36.8)	81/265 (30.6)
Cortex	128/500 (25.6)	73/231 (31.6)	5/269 (20.5)
PLIC	82/497 (16.5)	50/230 (21.7)	32/267 (12.0)
CC	63/460 (13.7)	35/227 (15.4)	28/233 (12.0)
Brainstem	44/501 (8.8)	31/232 (13.4)	13/269 (4.8)
Cerebellum	24/491 (4.9)	13/234 (5.6)	11/257 (4.3)

Abbreviations: BGT = basal ganglia thalami, CC = corpus callosum, PLIC = posterior-limb internal capsule, WM = white matter, D = day.

## Data Availability

Not applicable.

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
