# Peer review of "An Updated Overview of MRI Injuries in Neonatal Encephalopathy: LyTONEPAL Cohort"

_children, 2022, doi:10.3390/children9040561_

Round 1

Reviewer 1 Report

This article describes the characterization of brain injuries in neonatal encephalopathy based on MRI. The description is clear and includes important clinical messages.

  1. P3 line 98: Please state for abbreviation of PLIC in the main text before P9 line 195.
  2. P3 line 108: cMRI?
  3. For non-specialist readers of pediatrics, give a brief description of Sarnat & Sarnat grade.  
  4. In especially Table 2, please add the column of without “TH”.
  5. Please exemplify some MRI images of brain injuries in neonatal encephalopathy as a figure.

Author Response

Thank you for your remarks.

Here are our answer:

1: We added the statement for abbreviation PLIC as posterior limb internal capsule (P3, lines 101-102).

2: We suppressed the “c” before MRI (P3, line 117).

3: We added a short description of the Sarnat & Sarnat grade (P2, lines 94-96) “Grade of encephalopathy was evaluated by Sarnat & Sarnat score, which assesses consciousness, tone, reflexes, pupillary reactivity, oculomotricity, sucking and the presence/absence of seizures.”

4: We completed Table 2 with adding the column of newborn without TH and corrected an error concerning the injury rate of the vermis for those with TH (P5-6)

We suppressed the sentence below all the Tables: “Comparison of MRI between newborn with or without TH, by Fisher exact test for categorical variables”

5: We added some examples of MRI brain injuries. “Some examples of injury on cerebral MRI are presented on Figure 2.” (P6, lines 161-162)

And “Figure 2. MRI brain injuries in context of neonatal encephalopathy.” (P7, lines 166)

Reviewer 2 Report

The national cohort is representative, especially of the heterogeneity in neonatal HIE. However, the clinical impact of the current article needs to be strengthened.

Here are my questions:

Materials and Methods – Population

  • Were infants with brain injuries such as large subdural hemorrhage or subgaleal hemorrhage followed by systemic signs excluded from the study? Some infants showing depressed signs after birth were caused by traumatic birth process leading to skull fracture, SDH, or subgaleal hemorrhage, which could mimic HIE.
  • Are infants with sinovenous thrombosis or arterial ischemic stroke excluded from the cohort?

Results

  • Important information in Table 1 needs to be pointed out in the text, that could lead to a focus of discussion.
  • Table 2: If the MRI protocol is standardized, missing data should be very limited. Please explain why the denominators vary in each location.
  • Table 3: Case number of infants with TH should be verified.

Overall suggestions: Although infants in early-MRI group seems to be more severe clinically, dividing the patients into early-MRI and late-MRI groups may not bring about clinical impacts. The authors may compare the results with a historic cohort without TH, to highlight the changes of neuroimaging pattern in the era of hypothermia therapy.

Author Response

Thank you for your constructive remarks.

Here are our answers:

Materials and Methods- Population

  • We added this sentence in the methods (P2, lines 86-88): “Neonates with congenital malformations, chromosomal disorders and congenital neuromuscular disorders and traumatic birth without HIE criteria defined below were not included or were subsequently excluded.”
  • We added this sentence in the methods (P3, lines 111-114): “In another hand, radiologists were encouraged to describe the other types of injuries they observed other than those declined in the manuscript. Patients with isolated vascular lesions as sinus thrombosis or stroke were excluded. Traumatic injuries associated with hypoxic-ischemic criteria were considered if described by radiologist.”
  • We added this sentence in the discussion (page 11, lines 279-284): “Traumatic brain injuries have certainly been underestimated because of the design of the study.However, this time of lesions is often associated with instrumental maneuvers made necessary due the fetal conditions(fetal heart rhythm abnormalities). It remains particularly difficult in this context to distinguish the part due to traumatic birth injuries and hypoxic ischemic phenomenon, often associated in case of neonatal encephalopathy [35].”

Results:

  • Table 1.
    • We added this sentence (P5, lines 140-146): “Newborns under hypothermic condition were mainly born outside a TH center (72.5%). Of note, an acute perinatal event was observed for more than the half of the cohort (50,8%) and a fetal heart rate abnormality seen in a large part of the cohort (85 %). A large part of the cohort required intubation in delivery room (75,1%). Neonatal encephalopathy was moderate for the two third of the cohort (63.5% grade II). Newborns had a good short-term outcome with a large rate of normal neurological exam at discharge (72.4%). Death rate during hospitalization is low (15,6%).”
    • We commented these results in the beginning of the discussion (page 9-10, lines 206-210): “The characteristics of this present cohort are consistent with other cohorts dealing with neonatal encephalopathy due to hypoxic ischemic events: unexpected pathology, observation of perinatal events (sentinel events, fetal heart rhythm abnormalities, poor adaptation at the extra uterine life and a large proportion of mild encephalopathy) [5,15,16].”

  • Table 2:
    • We added the sentence in the discussion (page 11, lines 274-278): “However, despite an MRI standardized protocol and in absence of centralized review, every item has been not filled (explaining the variation of the denominators in Table 2) with an impact related to the quality of the MRI and the filling of the data by another person than the radiologist himself.”

  • Table 3:
    • We have corrected the second column of Table 3, the number of patients of 520 is correct, it concerned all newborn ≥ 36 wg and not only those on TH, we suppressed that error (P7-8).

Overall suggestions:

We added this sentence in the discussion (P10 , lines 238-241): “In comparison with previous studies, this present study highlights the positive impact of therapeutic hypothermia with an observation of an higher rate of normal MRI [3,4,18,27–30], a lower or stable rate of severe lesions as BGT [3,16,18,27–30] and a lower rate of cortical injuries [3,16,18,28–30].”

We have also added a conclusion to reinforce the clinical impact of this article, (P11, lines 287-295):“This study explored the panorama of brain injuries in the context of neonatal encephalopathy due to hypoxic-ischemic events. Basal ganglia injuries, especially involving thalami, and white matter injuries within the periventricular location were predominant. A comprehensive identification of brain injuries in the context of HIE may provide insight into involving mechanisms and their time of occurrence. Confrontation of anatomical injuries observed in the neonatal period remains necessary to improve prognostication aiming to deliver precise information for parents and also to personalize the future cares of the infant.”

Round 2

Reviewer 2 Report

The questions are answered with modifications in the article. I have no further query.